# IncEventGS: Pose-Free Gaussian Splatting from a Single Event Camera

## Abstract

Implicit neural representation and explicit 3D Gaussian Splatting (3D-GS) for novel view synthesis have achieved remarkable progress with frame-based camera (*e.g.* RGB and RGB-D cameras) recently. Compared to frame-based camera, a novel type of bio-inspired visual sensor, *i.e.* event camera, has demonstrated advantages in high temporal resolution, high dynamic range, low power consumption and low latency. Due to its unique asynchronous and irregular data capturing process, limited work has been proposed to apply neural representation or 3D Gaussian splatting for an event camera. In this work, we present *IncEventGS*, an incremental 3D Gaussian Splatting reconstruction algorithm with a single event camera. To recover the 3D scene representation incrementally, we exploit the tracking and mapping paradigm of conventional SLAM pipelines for *IncEventGS*. Given the incoming event stream, the tracker firstly estimates an initial camera motion based on prior reconstructed 3D-GS scene representation. The mapper then jointly refines both the 3D scene representation and camera motion based on the previously estimated motion trajectory from the tracker. The experimental results demonstrate that *IncEventGS* delivers superior performance compared to prior NeRF-based methods and other related baselines, even we do not have the ground-truth camera poses. Furthermore, our method can also deliver better performance compared to state-of-the-art event visual odometry methods in terms of camera motion estimation.

## 1 Introduction

Acquiring accurate 3D scene representations from 2D images has long been a challenging problem in computer vision. Serving as a fundamental component in various applications such as virtual/augmented reality and robotics navigation *etc.*, substantial efforts have been dedicated to addressing this challenge over the last few decades. Among those pioneering works, Neural Radiance Fields (NeRF) Mildenhall et al. (2021) and 3D Gaussian Splatting (3D-GS) Kerbl et al. (2023), stand out for its utilization of differentiable rendering technique, and have garnered significant attention due to its capability to recover high-quality 3D scene representation from 2D images. Commonly used sensors for 3D scene reconstruction are usually frame-based cameras, such as the RGB and RGB-D cameras. They usually capture full-brightness intensity images within a short exposure time at a regular frequency. Due to the characteristic of the data capturing process, they often suffer from motion blur or fail to capture accurate and informative intensity information under extreme brightness or darkness in the environment, which would further affect the performance of downstream applications.

The event camera, a bio-inspired sensor, has gained significant attention in recent years for its potential to address the limitations of frame-based cameras under challenging conditions. Unlike conventional cameras, event cameras record brightness changes asynchronously at each pixel, emitting events when a predefined threshold is surpassed. This unique operation offers several advantages over conventional cameras, in terms of high temporal resolution, high dynamic range, low latency and power consumption. Although event cameras have attractive characteristics for challenging environments, they cannot be directly integrated into existing frame-based 3D reconstruction algorithms that rely on processing dense 2D brightness intensity images, due to its time continuous, sparse, asynchronous and irregular data capturing characteristics.

Several pioneering works have been proposed to exploit event stream Kim et al. (2016); Rebecq et al. (2017); Gallego et al. (2018) to recover the motion trajectory and scene representation. While existing methods deliver impressive performance, they usually exploit 2.5D semi-dense depth maps to represent the 3D scene, and bundle adjustment (BA) is hardly being performed, due to the asynchronous and sparse characteristics of event data stream. Klenk et al. (2024) recently proposes to convert event stream into event voxel grids, and then adapt a previous frame-based deep visual odometry pipeline Teed et al. (2023) for accurate camera motion estimation. As neural radiance fields (NeRF) exhibits impressive scene representation capability recently, Klenk et al. (2022), Hwang et al. (2023), Rudnev et al. (2023a), Low & Lee (2023) and Low & Lee (2023) explore to recover the underlying dense 3D scene NeRF representation from event stream, by assuming ground truth poses are available.

In contrast to those works, we propose *IncEventGS*, an incremental dense 3D scene reconstruction algorithm from a single event camera, by exploiting Gaussian Splatting as the underlying scene representation. Different from prior event-based NeRF reconstruction methods, *IncEventGS* does not require any ground truth camera poses, which is more challenging and provides more flexibility for real-world application scenarios. To overcome the challenges brought by unknown poses, *IncEventGS* adopts the tracking and mapping paradigm of conventional SLAM pipelines Mur-Artal & Tardós (2017). In particular, *IncEventGS* exploits prior explored and reconstructed 3D scene for camera motion estimation of incoming event stream during the tracking stage. Both the 3D-GS scene representation and camera motions are then jointly optimized (*i.e.* event-based bundle adjustment) during the mapping stage, for more accurate scene representation and motion estimation. The 3D scene is progressively expanded and densified. Both synthetic and real datasets are used to evaluate our method. The experimental results demonstrate that *IncEventGS* is able to recover the underlying 3D scene representation and camera motion trajectory accurately. In particular, *IncEventGS* outperforms prior NeRF-based methods and other related baselines in terms of scene representation recovery, even *IncEventGS* does not have the ground-truth poses. Furthermore, our method also delivers better camera motion estimation accuracy than a most recent state-of-the-art visual odometry algorithm, in terms of both the Absolute Trajectory Error (ATE) metric. The recovered 3D scene representation can be further used to render novel brightness images. Our main contributions can be summarized as follows:

- We present an incremental 3D Gaussian Splatting reconstruction algorithm from a single event camera, without requiring the ground truth camera poses;
- The experimental results on both the synthetic and real datasets demonstrate superior performance of our method over prior NeRF based methods and related baselines in terms of novel view synthesis, and better performance over state-of-the-art event-based visual odometry algorithm in terms of camera motion estimation;
- Compared to prior methods, we are able to efficiently render high-quality brightness images thanks to the powerful representation capability of 3D Gaussian Splatting.

## 2 RELATED WORKS

We review two main areas of prior works: event-based neural radiance fields and 3D Gaussian Splatting, which are the most related to our work.

**Event-based neural radiance fields.** Prior works from Klenk et al. (2022), Hwang et al. (2023) and Rudnev et al. (2023a) propose to exploit event stream to recover the neural radiance fields with known camera motion trajectory. Low & Lee (2023) further improves the reconstruction algorithm to handle the situation with sparse and noisy events under non-uniform motion. The recovered neural radiance fields can then be used to render novel view brightness images. The ground truth poses are usually computed from corresponding brightness images via COLMAP Schonberger & Frahm (2016) or provided by indoor motion capturing system. Recently, Qu et al. (2024) proposed to integrate event measurements into an RGB-D implicit neural SLAM framework and achieve robust performance under the situation with motion blur. Li et al. (2024) also propose to exploit event measurements and a single blurry image to recover the underlying neural 3D scene representation. In contrast to those works, *IncEventGS* conduct incremental 3D scene reconstruction without requiring any prior ground truth poses, which is more challenging and provides more flexibility for practical application scenarios.

The method further exploits 3D Gaussian Splatting as the underlying scene representation, which demonstrates better image rendering quality and efficiency, compared to NeRF-based representation.

**3D Gaussian Splatting.** With the recent success of NeRF Mildenhall et al. (2021) and its further developments Fridovich-Keil et al. (2022); Müller et al. (2022), novel view synthesis utilizing 3D representations like MLPs, voxel grids, or hash tables has advanced significantly. While these NeRF-inspired models perform admirably, they frequently require long training and rendering times for individual scenes. The introduction of 3D Gaussian Splatting Kerbl et al. (2023) proposes a novel explicit 3D representation to further improve both the training and rendering efficiency. Due to its impressive efficient scene representation capability, several pioneering work have been proposed to exploit 3D-GS for incremental 3D reconstruction. For example, Keetha *et al.* propose an RGBD-based 3D-GS SLAM Keetha et al. (2024), employing an online tracking and mapping system tailored to the underlying Gaussian representation. Yan *et al.* implement a coarse-to-fine camera tracking approach based on the sparse selection of Gaussians Yan et al. (2024). Matsuki *et al.* propose to apply 3D Gaussian Splatting to do incremental 3D reconstruction using a single moving monocular or RGB-D camera Matsuki et al. (2024). Huang *et al.* exploits ORB-SLAM3 to compute accurate camera poses and feeds it into a 3D-GS algorithm for dense mapping Huang et al. (2024). Fu *et al.* uses monocular depth estimation with 3D-GS Fu et al. (2023). Yugay *et al.* combine DROID-SLAM Teed & Deng (2021) based camera tracking with active and inactive 3D-GS sub-maps Yugay et al. (2023). Hu *et al.* propose a novel depth uncertainty model to ensure the selection of valuable Gaussian primitives during optimization Hu et al. (2024). While those methods deliver impressive performance in terms of 3D scene recovery and motion estimation, they usually assume the usage of frame-based images (*i.e.* either RGB or RGB-D date). In the contrary, we propose to exploit pure event measurements for incremental 3D-GS reconstruction. Two concurrent work have also tried to exploit 3D Gaussians for event-based reconstruction recently, *i.e.* EvGGS Wang et al. (2024) and Event3DGS Xiong et al. (2024). However, both of them rely on ground-truth poses for training, and EvGGS further constrained to object reconstruction from 360-degree surrounding views, while ours assumes the camera poses are not available and is more challenging.

## 3 METHOD

The overview of our *IncEventGS* is shown in Fig. 1. Given only a single event camera, *IncEventGS* incrementally performs tracking and dense mapping under the framework of 3D Gaussian Splatting, to recover both the camera motion trajectory and 3D scene representation simultaneously. The main insight of *IncEventGS* is to accumulate incoming event data into chunks and treat each chunk as a special "image". We associate each chunk with a continuous time trajectory parameterization in the $\mathfrak{se}3$ space. Two close consecutive timestamps (i.e., $t_k$ and $t_{k+\Delta t}$, where $\Delta t$ is a small time interval) can be randomly sampled within the chunk and the corresponding event stream can then be integrated into an image $\mathbf{E}(x)$. Based on the parameterized trajectory, the corresponding camera poses (*i.e.*, $\mathbf{T}_k$, $\mathbf{T}_{k+\Delta t}$) can be computed and the images (*i.e.*, $\hat{\mathbf{I}}_k$, $\hat{\mathbf{I}}_{k+\Delta t}$) can be further rendered from the 3D Gaussian Splatting. The synthesized image $\hat{\mathbf{E}}(x)$ can be computed for event loss computation. During tracking, we only optimize for the camera motion trajectory of the newly accumulated event chunk and exploit the recovered trajectory to initialize the dense bundle adjustment (BA) algorithm for the mapping stage. During mapping stage, we continuously densify 3D Gaussians for newly explored areas and prune transparent 3D Gaussians. For computational efficiency consideration, we exploit a sliding window of the latest chunks and perform BA only within this window for both 3D-GS reconstruction and motion trajectory estimation. We will detail each component as follows.

### 3.1 3D SCENE REPRESENTATION

Following 3D-GS Kerbl et al. (2023), the scene is represented by a set of 3D Gaussian primitives, each of which contains mean position $\boldsymbol{\mu} \in \mathbb{R}^3$ in the world coordinate, 3D covariance $\boldsymbol{\Sigma} \in \mathbb{R}^{3\times3}$, opacity $\mathbf{o} \in \mathbb{R}$, and color $\mathbf{c} \in \mathbb{R}^3$. To ensure that the covariance matrix remains positive semi-definite throughout the gradient descent, the covariance $\boldsymbol{\Sigma}$ is parameterized using a scale vector $\mathbf{s} \in \mathbb{R}^3$ and rotation matrix $\mathbf{R} \in \mathbb{R}^{3\times3}$:

$$\boldsymbol{\Sigma} = \mathbf{R}\mathbf{S}\mathbf{S}^T\mathbf{R}^T, \tag{1}$$

where scale matrix $\mathbf{S} = diag([s])$ is derived from the scale vector $\mathbf{s} \in \mathbb{R}^3$.

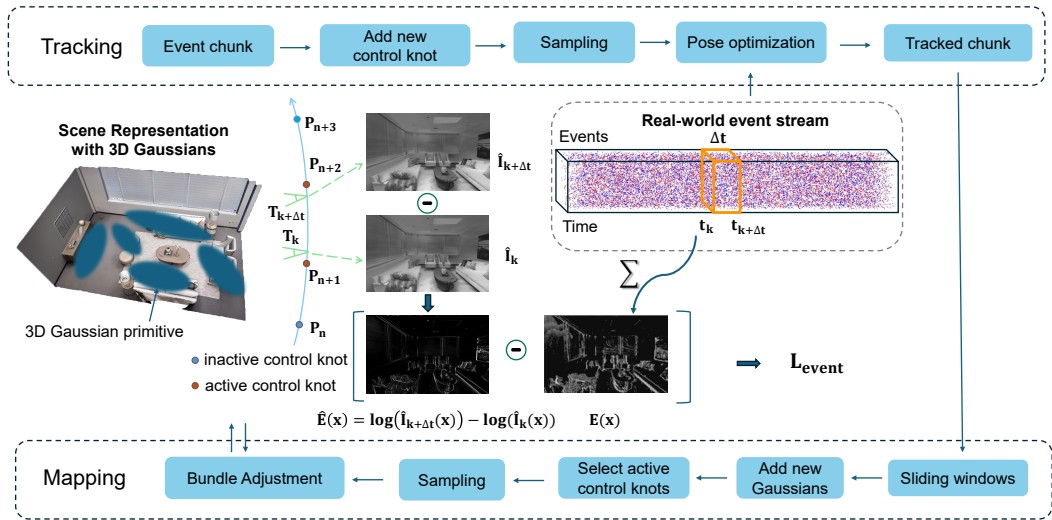

Figure 1: **The pipeline of *IncEventGS*.** *IncEventGS* consists of a camera motion tracker and a sliding window dense mapper based on the 3D-GS scene representation. It divides incoming event stream data into event chunks and uses a continuous time representation for the corresponding motion trajectory. During the tracking stage, we optimize the trajectory based on prior recovered 3D-GS representation and the real event measurements. The event chunk with estimated motion trajectory is then inserted into the mapper. Both the 3D-GS and the motion trajectory are then jointly optimized by exploiting the event data formation process. The tracker and the mapper are optimized alternatively as the event stream is continuously fed into the pipeline.

In order to enable rendering, 3D-GS projects 3D Gaussian primitives to the 2D image plane from a given camera pose $\mathbf{T}_c = \left\{ \mathbf{R}_c \in \mathbb{R}^{3\times3}, \mathbf{t}_c \in \mathbb{R}^3 \right\}$ using following equation:

$$\Sigma' = \mathbf{J}\mathbf{R}_c\Sigma\mathbf{R}_c^T\mathbf{J}^T, \tag{2}$$

where $\Sigma' \in \mathbb{R}^{3\times3}$ is the 2D covariance matrix, $\boldsymbol{J} \in \mathbb{R}^{2\times3}$ is the Jacobian of the affine approximation of the projective transformation. After projecting 3D Gaussians onto the image plane, the color of each pixel is determined by sorting the Gaussians according to their depth and then applying near-to-far $\alpha$-blending rendering via the following equation:

$$\mathbf{I} = \sum_{i}^{N} \mathbf{c}_i\alpha_i \prod_{j}^{i-1} (1 - \alpha_j), \tag{3}$$

where $\mathbf{c}_i$ is the learnable color of each Gaussian, and $\alpha_i$ is the alpha value computed by evaluating the 2D covariance $\Sigma'$ multiplied with the learned Gaussian opacity $\mathbf{o}$:

$$\alpha_i = \mathbf{o}_i \cdot \exp\left(-\sigma_i\right), \quad \sigma_i = \frac{1}{2}\Delta_i^T\Sigma'^{-1}\Delta_i, \tag{4}$$

where $\Delta_i \in \mathbb{R}^2$ is the offset between the pixel center and the 2D Gaussian center. Depth is rendered by:

$$\mathbf{D} = \sum_{i}^{N} \mathbf{d}_i\alpha_i \prod_{j}^{i-1} (1 - \alpha_j), \tag{5}$$

where $d_i$ denotes the z-depth of the center of the i-th 3D Gaussian to the camera. We also render alpha map to determine visibility:

$$\mathbf{V} = \sum_{i}^{N} \alpha_i \prod_{j}^{i-1} (1 - \alpha_j), \tag{6}$$

The derivations presented above demonstrate that the rendered pixel color, denoted as C in (3), is a function that is differentiable with respect to the learnable attributes of all Gaussians, and the camera poses $\mathbf{T}_c$. This facilitates our bundle adjustment formulation, accommodating a set of event chunks and inaccurate camera motion trajectories within the framework of 3D-GS.

## 3.2 Event Data Formation Model

An event camera records changes of the brightness as a stream of events asynchronously. Every time a pixel brightness change reaches a contrast threshold (*i.e.* $|L(\mathbf{x}, t_i + \delta t) - L(\mathbf{x}, t_i)| \geq C$), the camera will trigger an event $e_i = (\mathbf{x}, t_i, p_i)$, where $p_i \in (-1, +1)$ is the polarity of the event, $L(\mathbf{x}, t_i) = \log(\mathbf{I}(\mathbf{x}, t_i))$ is the brightness logarithm of pixel $\mathbf{x}$ at timestamp $t_i$, $C$ is the fixed contrast threshold.

To relate 3D-GS representation with the event stream, we sample two close consecutive timestamps (i.e., $t_k$ and $t_{k+\Delta t}$, where $\Delta t$ is a small time interval), and accumulate the real measured events within $\Delta t$ to an image $\mathbf{E}(\mathbf{x})$. The accumulation is defined as:

$$\mathbf{E}(\mathbf{x}) = C\{e_i(\mathbf{x}, t_i, p_i)\}_{t_k < t_i < t_k + \Delta t}, \tag{7}$$

where $e(\mathbf{x}, t_i, p_i)$ is the $i^{th}$ event within the defined time interval corresponding to pixel $\mathbf{x}$. The corresponding camera poses $T_k$ and $T_{k+\Delta t}$ can be interpolated from the camera motion trajectory parameterization, allowing us to render two grayscale images (*i.e.* $\hat{\mathbf{I}}_k$ and $\hat{\mathbf{I}}_{k+\Delta t}$) from the previously recovered 3D-GS. The synthesized accumulated event image $\hat{\mathbf{E}}$ can then be computed as:

$$\hat{\mathbf{E}}(\mathbf{x}) = \log(\hat{\mathbf{I}}_{k+\Delta t}(\mathbf{x})) - \log(\hat{\mathbf{I}}_k(\mathbf{x})), \tag{8}$$

where $\hat{\mathbf{E}}(\mathbf{x})$ depends on the parameters of both the motion trajectory parameters and 3D-GS, and is differentiable with respect to them.

Both in tracking and mapping, inspired by the work of Rudnev et al. (2023b), we segment the current event chunks into $n_{seg}$ equal segments according to the number of events, obtaining $n_{seg}$ timestamps that correspond to the end of each segment. We then randomly select one timestamp from these $n_{seg}$ timestamps to serve as $t_{k+\Delta t}$, and we randomly sample an integer $n_{win}$ between the integer bounds $n_{low}$ and $n_{up}$. The index of $t_k$ is equal to the index of $t_{k+\Delta t}$ subtract $n_{win}$. $n_{seg}$, $n_{low}$ and $n_{up}$ are hyperparameters. This sampling strategy enables the model to capture both local and global information.

## 3.3 Camera Motion Trajectory Modeling

Since each event chunk usually contains too many events, we sample a portion of them according to the total number of events during optimization. We formulate the corresponding poses (*i.e.* $\mathbf{T}_k$ and $\mathbf{T}_{k+\Delta t}$) at the beginning and end of the sampled event portion within each chunk, by employing a camera motion trajectory. The trajectory is represented through linear interpolation between two camera poses, one at the beginning of the chunk $\mathbf{T}_{\text{start}} \in \mathbf{SE}(3)$ and the other at the end $\mathbf{T}_{\text{end}} \in \mathbf{SE}(3)$. The camera pose at time $t_k$ can thus be expressed as follows:

$$\mathbf{T}_k = \mathbf{T}_{\text{start}} \cdot \exp(\frac{t_k - t_{start}}{t_{end} - t_{start}} \cdot \log(\mathbf{T}_{\text{start}}^{-1} \cdot \mathbf{T}_{\text{end}})), \tag{9}$$

where $t_{start}$ and $t_{end}$ represent the timestamps corresponding to the boundary of the event chunk. It follows that $\mathbf{T}_k$ is differentiable with respect to both $\mathbf{T}_{\text{start}}$ and $\mathbf{T}_{\text{end}}$. The objective of *IncEventGS* is thus to estimate both $\mathbf{T}_{\text{start}}$ and $\mathbf{T}_{\text{end}}$ for each event chunk, along with the learnable parameters of 3D Gaussians $\mathbf{G}_\theta$.

## 3.4 Incremental Tracking and Mapping

For both tracking and mapping, we exploit the previously introduced event data formation model to compute the loss from the synthesized and real accumulated event images. In particular, we compute the loss of the latest event chunk only for the tracking stage and minimize the following energy function:

$$\mathbf{T}_{start}^*, \mathbf{T}_{end}^* = \underset{\mathbf{T}_{start}, \mathbf{T}_{end}}{\arg\min} \left\| \mathbf{E}(\mathbf{x}) - \hat{\mathbf{E}}(\mathbf{x}) \right\|_2, \tag{10}$$

where both $\mathbf{E}(\mathbf{x})$ and $\hat{\mathbf{E}}(\mathbf{x})$ are the accumulated real and synthesized event images respectively, corresponding to a randomly sampled event portion within the latest event chunk.

Once the tracking is done, we insert the latest event chunk to mapper and exploit the estimated $\mathbf{T}^*_{start}$ and $\mathbf{T}^*_{end}$ as the initial value of the chunk to perform dense bundle adjustment. For computational consideration, we exploit a sliding window BA of the latest $n_w$ chunks and $n_w$ is a hyperparameter. In particular, we optimize both the motion trajectories and the 3D-GS jointly by minimizing the following loss functions:

$$\mathcal{L} = (1 - \lambda)\mathcal{L}_{event} + \lambda \mathcal{L}_{ssim}, \tag{11}$$

$$\mathcal{L}_{event} = \frac{1}{n_w} \sum_{i=0}^{n_w} \left\| \mathbf{E}_i(\mathbf{x}) - \hat{\mathbf{E}}_i(\mathbf{x}) \right\|_2, \tag{12}$$

$$\mathcal{L}_{ssim} = \frac{1}{n_w} \sum_{i=0}^{n_w} SSIM(\mathbf{E}_i(\mathbf{x}), \hat{\mathbf{E}}_i(\mathbf{x})) \tag{13}$$

where $\lambda$ is a hyperparameter, SSIM is the structural dissimilarity loss Wang et al. (2004), both $\mathbf{E}_i(\mathbf{x})$ and $\hat{\mathbf{E}}_i(\mathbf{x})$ are the corresponding accumulated real and synthesized event images of the latest $i^{th}$ event portion respectively. As the event data streams in, we alternatively perform tracking and mapping.

**3D-GS Initialization and System Boot-strapping.** We initialize the 3D-GS by sampling point cloud randomly within a bounding box. The first $m$ event chunks (where $m$ is a hyperparameter) are selected for initialization, and all corresponding camera poses (e.g., $\mathbf{T}_*$) are randomly initialized to be near the identity matrix. We then minimize the loss computed by Eq. (11) with respect to the attributes of 3D-GS and the parameters of camera motion trajectories jointly.

Through experiments, we found that the above initialization procedure consistently produces high-quality brightness images. However, the 3D structure remains of low quality due to the short baselines of the event chunks. We further find that it could potentially affect the performance of the whole pipeline as more event data is received. Therefore, we utilize a monocular depth estimation network Ke et al. (2024) to predict a dense depth map from the rendered brightness image. This depth map is then used to re-initialize the centers of the 3D Gaussians by unprojecting the pixel depths, after which we repeat the minimization of Eq. (11) for system bootstrapping.

**3D-GS Incrementally Growing.** As the camera moves, new Gaussians is periodically introduced to cover newly explored regions. After tracking, we obtain an accurate camera pose estimate for each new event chunk. The center of new Gaussians are determined by:

$$p = T \cdot \pi^{-1}(u, d_u) \tag{14}$$

where $p \in \mathbb{R}^3$ is a 3D point, $u \in \mathbb{R}^2$ is a point in the image plane, $d_u$ is depth of the 3D point $p$ projecting on pixel u, which is rendered by equation 6, $\pi^{-1}$ denotes camera inverse projection, $T$ is the camera pose from tracking. To ensure that new Gaussians are only added in previously unmapped areas, a visibility mask is generated to guide the expansion of the Gaussian splatting process, as following:

$$M(p) = V < \lambda_V \tag{15}$$

where $V$ is the rendered alpha map, $\lambda_V$ is the hyperparameter.

## 4 EXPERIMENTS

### 4.1 EXPERIMENTAL SETUPS.

**Implementation Details.** All experiments were conducted on a desktop PC equipped with a 5.73GHz AMD Ryzen 9 7900x CPU and an NVIDIA RTX 3090 GPU. The first $m = 3$ event chunks were used for initialization. During the mapping stage, a sliding window size of $n_w = 20$ was employed for the bundle adjustment algorithm. The hyperparameters were set as follows: $\lambda = 0.05$, $\lambda_V = 0.8$, and $n_{seg} = 100$. For the synthetic dataset, $n_{low} = 400k$ and $n_{up} = 500k$, while for the real dataset, $n_{low} = 60k$ and $n_{up} = 80k$. Each event chunk had a time interval of 50 ms. The learning rate of the camera poses is set to 1e-4 and that for the attributes of 3D-GS are set the same as

the original 3D-GS work. The number of optimization steps for initialization is 4500, and that for tracking and mapping are set to 200 and 1500 respectively. The contrast threshold $C$ of the event camera is set to 0.1 for synthetic datasets and 0.2 for real datasets empirically.

**Baselines and Evaluation Metrics.** *IncEventGS* performs incremental dense 3D-GS reconstruction and motion estimation using only a single event camera. To the best of our knowledge, there are no existing event-only NeRF or 3D-GS SLAM methods that do not rely on ground-truth poses, making direct comparisons challenging. Therefore, we conduct a thorough comparison of our method with several event-based NeRF approaches, including E-NeRF Klenk et al. (2022), EventNeRF Rudnev et al. (2023a), and Robust e-NeRF Low & Lee (2023), as well as two-stage methods such as E2VID Rebecq et al. (2019) + COLMAP Schönberger & Frahm (2016) + 3DGS Kerbl et al. (2023), and E2VID Rebecq et al. (2019) + DEVO Klenk et al. (2024) + 3DGS Kerbl et al. (2023). E-NeRF, EventNeRF, and Robust e-NeRF leverage implicit neural radiance fields for 3D scene representation, requiring ground-truth camera poses for accurate NeRF reconstruction. The two-stage methods we examine include E2VID + COLMAP + 3DGS and E2VID + DEVO + 3DGS. In those approaches, event data is first converted into grayscale images using E2VID. The subsequent steps differ: in the E2VID + COLMAP + 3DGS method, camera poses are estimated from these images using COLMAP, while in the E2VID + DEVO + 3DGS method, poses are estimated using DEVO. Finally, 3D-GS is trained with the generated images and poses in both methods. Both the quantitative and qualitative comparisons are performed on the synthetic dataset. Since there are no paired ground truth images for the real dataset, we only perform qualitative comparisons on the real dataset. In terms of motion trajectory evaluations, we use the publicly available state-of-the-art event-only visual odometry method, *i.e.* DEVO Klenk et al. (2024), for comparison, both quantitatively and qualitatively.

The metrics used for novel view synthesis (NVS) include the commonly employed PSNR, SSIM, and LPIPS. For motion trajectory evaluations, we utilize Absolute Trajectory Error (ATE). To ensure fair comparisons, we employ the evaluation code provided by EventNeRF to compute the NVS metrics, which applies a linear color transformation between predictions and ground truth. Additionally, we use the public EVO toolbox Grupp (2017) to compute the trajectory metrics.

**Benchmark Datasets.** To properly evaluate the performance of NVS and motion trajectory estimation, we synthesized event data using the 3D scene models from the Replica dataset Straub et al. (2019). In particular, we exploit the *room0*, *room2*, *office0*, *office2*, and *office3* scenes. We rendered high frame rate RGB images at 1000 Hz with a resolution of 768x480 pixels. Those images are then converted to grayscale and the event data is generated via the events simulator Gehrig et al. (2020). The contrast threshold is set to 0.1. To simulate real-world camera motions, we exploit the same motion trajectories as that of NICE-SLAM Zhu et al. (2022) for data generation.

We used the event dataset provided by TUM-VIE Klenk et al. (2021) for real data evaluations, which is also used by E-NeRF and Robust e-NeRF. TUM-VIE Klenk et al. (2021) captured the event datasets by a pair of Prophesee Gen4 HD event cameras with a resolution of 1280x720 pixels. We only use the left-event camera data for our experiment.

### 4.2 QUANTITATIVE EVALUATIONS.

We conduct quantitative evaluations against event NeRF methods(E-NeRF, EventNeRF, and Robust e-NeRF) and two-stage methods (E2VID + COLMAP + 3DGS and E2VID + DEVO + 3DGS) in terms of the quality of NVS and pose estimation performance.

The NVS performance is evaluated on Replica-dataset and the results are presented in Table 1. It is important to note that the metrics are lower than those typically observed in standard NeRF/3D-GS methods for RGB images, primarily due to the absence of adequate RGB image supervision. Even though nerf-based methods use ground truth poses for training, *IncEventGS* still significantly outperforms them, highlighting the advantages of our approach utilizing a 3D Gaussian representation. Additionally, our method greatly surpasses two-stage methods that also employ 3D Gaussian representation, demonstrating superior pose estimation and the effectiveness of our bundle adjustment technique.

We evaluate pose estimation performance using the $ATE$ metric on both synthetic and real datasets, comparing our method with EVO and E2VID + COLMAP. The results, presented in Table 2, show

Table 1: NVS performance comparison on Replica dataset. The result demonstrates that our method outperforms NeRF-based and two-stage methods.

| | room0 | | | room2 | | | office0 | | | office2 | | | office3 | | |
|---|---|---|---|---|---|---|---|---|---|---|---|---|---|---|---|
| | PSNR↑ | SSIM↑ | LPIPS↓ | PSNR↑ | SSIM↑ | LPIPS↓ | PSNR↑ | SSIM↑ | LPIPS↓ | PSNR↑ | SSIM↑ | LPIPS↓ | PSNR↑ | SSIM↑ | LPIPS↓ |
| E-NeRF | 13.99 | 0.58 | 0.51 | 15.56 | 0.47 | 0.58 | 18.91 | 0.51 | 0.57 | 13.05 | 0.65 | 0.44 | 14.01 | 0.62 | 0.48 |
| EventNeRF | 17.29 | 0.62 | 0.39 | 16.02 | 0.54 | 0.64 | 18.90 | 0.43 | 0.62 | 15.18 | 0.66 | 0.45 | 16.77 | 0.73 | 0.33 |
| Robust e-NeRF | 17.26 | 0.84 | 0.18 | 16.43 | 0.50 | 0.52 | 18.93 | 0.52 | 0.56 | 16.81 | 0.81 | 0.25 | 19.22 | 0.84 | 0.18 |
| E2VID+ COLMAP+3DGS | 14.45 | 0.44 | 0.52 | 15.74 | 0.51 | 0.55 | 18.91 | 0.31 | 0.68 | 14.03 | 0.57 | 0.48 | 13.25 | 0.47 | 0.53 |
| E2VID+ DEVO+3DGS | 14.35 | 0.42 | 0.56 | 15.73 | 0.49 | 0.59 | 18.90 | 0.28 | 0.72 | 13.84 | 0.56 | 0.55 | 13.28 | 0.46 | 0.57 |
| Ours | **24.31** | **0.85** | **0.17** | **23.75** | **0.79** | **0.23** | **25.64** | **0.54** | **0.30** | **21.74** | **0.82** | **0.23** | **21.18** | **0.88** | **0.13** |

Table 2: Pose accuracy (ATE, cm) on Replica and TUM-VIE datasets. The results demonstrate that our method delivers better performance in terms of camera motion estimation.

| | room0 | room2 | office0 | office2 | office3 | 1d | 3d | 6dof | desk | desk2 |
|---|---|---|---|---|---|---|---|---|---|---|
| DEVO | 0.289 | 0.266 | 0.138 | 0.281 | 0.156 | 0.147 | 0.303 | 2.93 | 0.732 | 0.201 |
| E2VID+COLMAP | 17.93 | 59.96 | 105.19 | 18.414 | 17.28 | 4.268 | 16.90 | 9.88 | 21.57 | 10.13 |
| Ours | **0.046** | **0.067** | **0.045** | **0.046** | **0.054** | **0.115** | **0.298** | **0.251** | **0.231** | **0.129** |

that our method outperforms both baselines, validating the effectiveness of our incremental tracking and mapping technique.

## 4.3 QUALITATIVE EVALUATIONS.

We evaluate our method against event NeRF methods and two-stage methods qualitatively in terms of novel view image synthesis, both on synthetic and real data. The results are presented in both Fig. 2 and Fig. 3. It demonstrates that our method can deliver better novel view images, while event NeRF methods and two-stage methods render images with additional artifacts. Compared to NeRF-based methods, our approach demonstrates the advantage of *IncEventGS* by leveraging 3D Gaussian Splatting as the underlying scene representation. In contrast to two-stage methods, our dense bundle adjustment optimizes both 3D Gaussian Splatting and camera pose using event data, whereas two-stage approaches tend to accumulate errors over time, as confirmed by the experimental results. We also provide representative visualization of ATE error mapped onto trajectories in Fig. 4, both on synthetic and real dataset. It demonstrates that *IncEventGS* is able to recover more accurate motion trajectories.

## 4.4 ABLATION STUDY

We conduct ablation studies to confirm our design choices. In particular, we study the effect of a monocular depth estimation network for system bootstrapping and event slicing hyperparameters $n_{low}, n_{up}$. The experiments are conducted with the Replica dataset and the results are shown in Table 3 and Table 4 respectively.

Table 3: Ablation Study about Depth Initialization. The unit of ATE is cm. The experimental results demonstrate the effectiveness of the initialization strategy. It not only improves the quality of rendered images, but also improves the accuracy of the camera motion estimation significantly.

| Setting | PSNR↑ | SSIM↑ | LPIPS↓ | ATE |
|---|---|---|---|---|
| full | **21.74** | **0.82** | **0.23** | **0.046** |
| w/o | 17.80 | 0.76 | 0.26 | 1.534 |

Table 4: Ablation Study on Event Slice Window Size (Hyperparameters $n_{low}$ and $n_{up}$). The unit of ATE is cm.

| Setting | PSNR↑ | SSIM↑ | LPIPS↓ | ATE |
|---|---|---|---|---|
| 1k-10k | 16.07 | 0.64 | 0.46 | 0.167 |
| 10k-50k | 18.41 | 0.72 | 0.33 | 0.079 |
| 80k-200k | 20.99 | 0.79 | 0.25 | 0.079 |
| **400k-500k** | **21.74** | **0.82** | **0.23** | **0.046** |
| 500k-600k | 20.95 | 0.79 | 0.23 | 0.050 |
| 600k-700k | 18.06 | 0.75 | 0.28 | 0.214 |

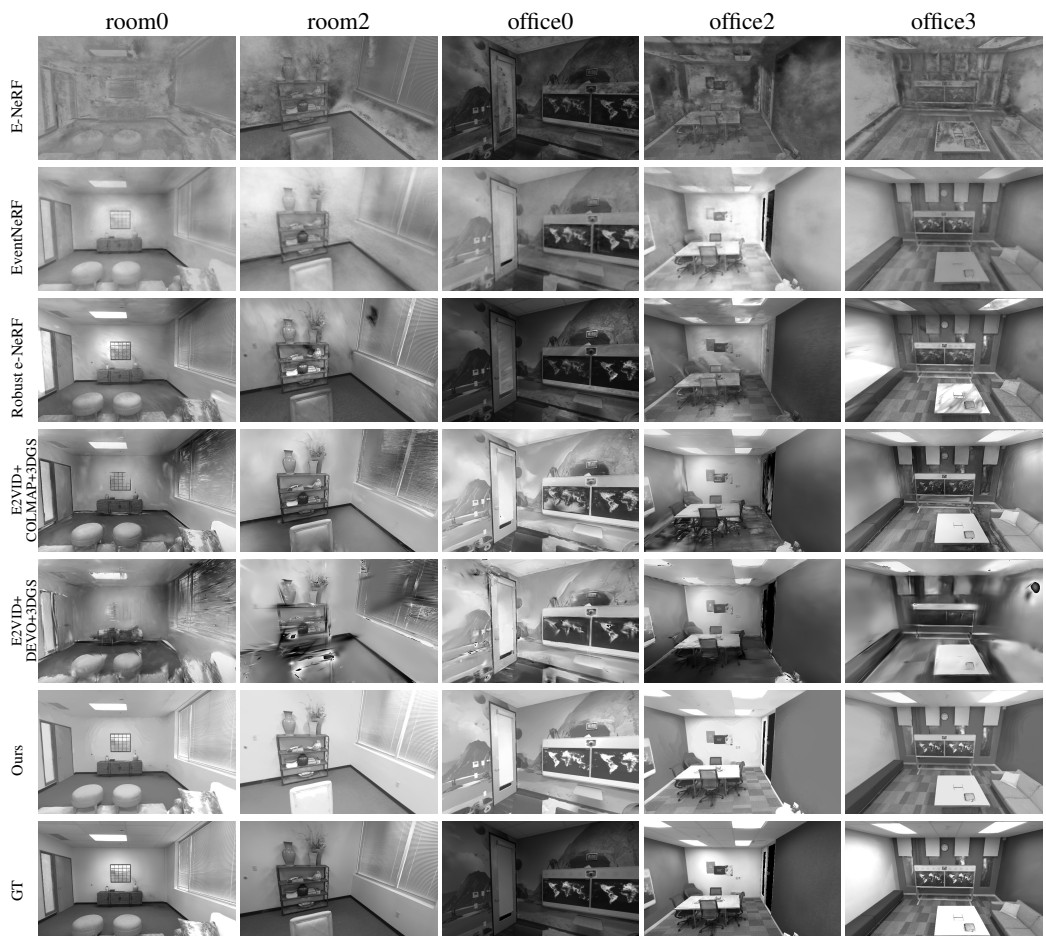

Figure 2: Qualitative evaluation of novel view image synthesis on the Replica dataset. The experimental results demonstrate that our method renders higher-quality images with fewer artifacts compared to event-based NeRF and two-stage approaches.

We found that depth initialization significantly impacts pose estimation, reducing the Average Trajectory Error (ATE) from 1.534 cm to 0.064 cm. Additionally, this improvement in pose estimation leads to a slight enhancement in Novel View Synthesis (NVS) performance. These results verify the importance of using depth initialization.

We compare several combinations of hyperparameters $n_{low}$ and $n_{up}$, which refer to the range of event slicing window sizes. Table 4 demonstrates that both too small and too large window sizes negatively impact the performance of Novel View Synthesis (NVS) and pose estimation. Consequently, we select $n_{low} = 400k$ and $n_{up} = 500k$ for our experiments on the Replica dataset.

## 5 CONCLUSION

We present the first incremental 3D dense reconstruction algorithm, *i.e. IncEventGS*, with a single event camera under the framework of 3D Gaussian Splatting. We adopt the tracking and mapping paradigm in conventional SLAM pipeline to do incremental motion estimation and 3D scene reconstruction simultaneously. To handle the continuous and asynchronous characteristics of event stream, we exploit a continuous trajectory model to model the event data formation process. The experimental results on both synthetic and real datasets demonstrate the superior performance of *IncEventGS* over prior state-of-the-art methods in terms of high-quality novel image synthesis and camera pose estimation.

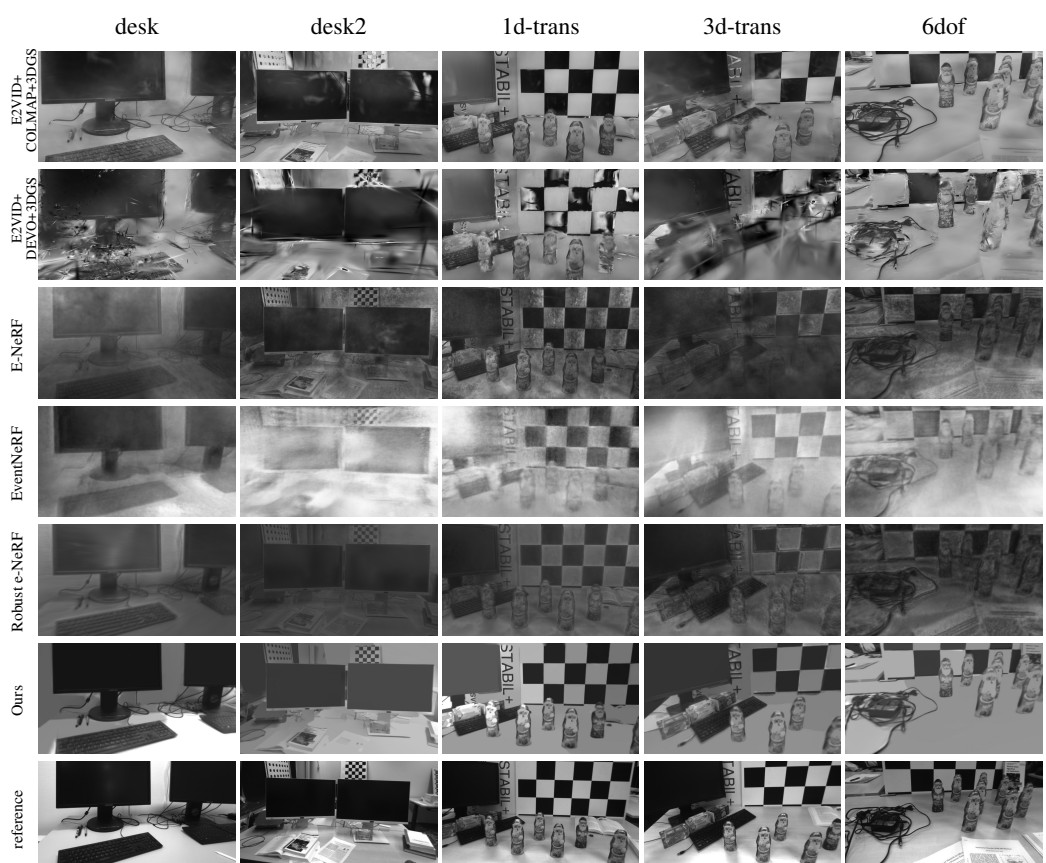

Figure 3: Qualitative evaluation for novel view image synthesis on real dataset. It demonstrates that our method is able to render better images with fewer artifacts than event NeRF methods and two-stage methods. Note there are no GT images aligned with the event camera, we choose closest images of RGB camera and crop it to the the size with rendered images for visual comparisons.

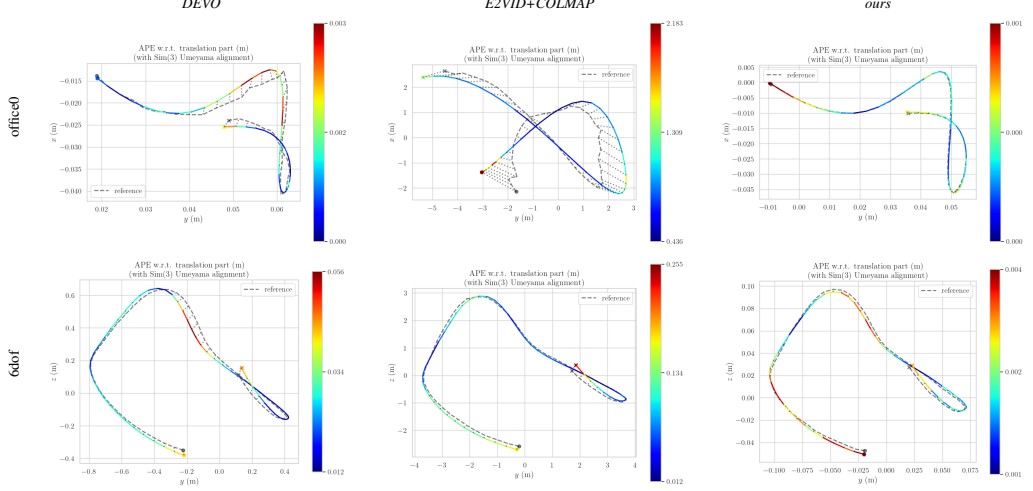

Figure 4: Representative visualization of ATE error mapped onto trajectories for the synthetic (office0) and real (6dof) datasets, generated by the EVO toolbox using the same ground truth poses, demonstrating the superior performance of our method in pose estimation.

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
