# OpenReview forum: "IncEventGS: Pose-Free Gaussian Splatting from a Single Event Camera"
_ICLR.cc/2025/Conference — ICLR 2025 Conference Withdrawn Submission_

### Official Review · Reviewer_xHif · 2024-10-15

**Soundness:** 3
**Presentation:** 2
**Contribution:** 3
**Rating:** 3
**Confidence:** 5

**Summary:**

The present paper proposes a novel view synthesis approach that reconstructs 3D scenes from event camera streams without precise camera poses. The main goal is to handle the unique asynchronous and irregular data characteristics of event cameras, which pose significant challenges for traditional 3D reconstruction methods. By utilizing a tracking and mapping approach inspired by SLAM pipelines, the method estimates camera motion based on prior reconstructed 3D-GS scenes and incrementally refines both the camera motion and scene representation. It's capable of handling real-world scenarios with no gt. poses, offering improved performance compared to NeRF-based methods and event-based visual odometry. It efficiently renders high-quality brightness images, outperforming baseline methods in terms of novel view synthesis and motion estimation.

**Strengths:**

Overall, this paper presents an incremental 3D Gaussian Splatting reconstruction algorithm from a single event camera, without requiring the ground truth camera poses. It has a motivation and is adequate for the audience and also solid on the technical side and adds the event-based VO tricks and off-the-shelf depth model for re-initialization to 3DGS. Thus, this work is interesting for readers working at the intersection of novel view synthesis and neuromorphic vision.

**Weaknesses:**

1. The setup of the real-world experiments lacks validity. Specifically, for the evaluation on the TUM-VIE dataset, the qualitative results alone are insufficient. The authors should also include quantitative analysis using no-reference image quality assessment metrics to provide a more comprehensive evaluation.

2. Although the proposed method can operate using an event stream as input to reconstruct 3D Gaussians, it still relies on uniform event stream as input. The proposed method is, therefore, limited by the density of event data streams, which restricts its practical applications.

3. Despite the detailed comparison of the quality of rendered images, the efficiency of the training and rendering process is not included, which is an important metric of NVS methods. Extra comparisons with other methods on training time and inference FPS would help better evaluate the proposed method.

4. This method is valuable for addressing event-based visual odometry. However, the authors focus more on the NVS task, and using Gaussian functions to reconstruct grayscale scenes seems less relevant, as they are mainly suited for head-mounted devices, which reduces the method’s rationale.

Beyond this I have mainly minor comments and nitpicks:

l.117, the sentence contains a grammatical error and should be revised. Specifically, "IncEventGS conduct..." should be corrected to "IncEventGS conducts...".

l.142, the expression should be standardized by changing "se3" to "se(3)" for clarity and consistency.

l.162~186, I think the re-initialization process is vital to the method, but the main figure of the pipeline does not reflect this which may generate some confusion with readers not familiar with the method.

**Questions:**

1. The re-initialization using the pre-trained depth model for regularization with the proposed Gaussian model is not clear. Can the authors provide more details about it? Especially regarding the visualization of the intermediate process.

2. For SLAM or VIO, the accuracy of the trajectory is crucial. However, for NVS (Novel View Synthesis) tasks, the proposed method merely reconstructing a gray map of the scene can diminish the significance of the task to some extent. It is not enough to work only on the gray map. Could we perform the NVS task on the RGB event dataset? For example, the dataset from [1] or [2] or the event-based color synthetic Replica dataset.

3. What's more, I noticed that the authors did not provide any supplementary materials. Could the authors provide some visual demos to better observe the overall effect of this method?

[1] Robust e-NeRF: NeRF from Sparse & Noisy Events under Non-Uniform Motion

[2] EventNeRF: Neural Radiance Fields from a Single Color Event Camera

---

### Official Review · Reviewer_gT1i · 2024-11-02

**Soundness:** 4
**Presentation:** 2
**Contribution:** 3
**Rating:** 5
**Confidence:** 4

**Summary:**

This paper proposes IncEventGS, which is an incremental 3D Gaussian Splatting reconstruction algorithm with a single event camera. IncEventGS employed the tracking and mapping paradigm of conventional SLAM pipelines.

**Strengths:**

1.  The results of IncEventGS shown in Tabs 1 and 2  are amazing and effective.

2.  This paper is written in an easy-to-understand manner.

**Weaknesses:**

1. The motivation of this paper is weak, as the paper claims, "Due to its unique asynchronous and irregular data capturing process, limited work has been proposed to apply neural representation or 3D Gaussian splatting for an event camera". I think the authors should discover the reasons behind it rather than the superficial phenomenon.

2. The title is "Pose-free." Why the author did this is not explained.  I think that although no pose ground truth is provided, using conventional slam pipelines actually provides this variable implicitly. Conventional slam pipelines will be more robust than pose estimators, which use deep learning methods.

3. This paper mentions several times “due to its time continuous, sparse, asynchronous and irregular data capturing characteristics.” I don't think the authors have solved this problem; they are still taking the approach of stacking events into the frame.

4. In line 62, citation duplication.

5. The contribution needs to be rewritten, which is just like changing the representation from Nerf to GS. However, this work has already been done.

6. "The main insight of IncEventGS is to accumulate incoming event data into chunks and treat each chunk as a special "image".  This is not a contribution and does not need to be emphasized.

7. In line 216 and 307, C in (3) and equation 6.

**Questions:**

1. Why did IncEventGS stop using the ground-truth pose after adopting Gaussian Splatting representations compared to Nerf-based representations?

2. As we know, 3DGS hardware friendliness is superior to Nerf-based representations, and I'm curious about the overall runtime of the system compared to Nerf-based.

3. More experiments need to be done, such as  Tanks and Temples.

---

### Official Review · Reviewer_UfLZ · 2024-11-03

**Soundness:** 2
**Presentation:** 2
**Contribution:** 3
**Rating:** 5
**Confidence:** 4

**Summary:**

1) This manuscript presents a method in which a 3D scene is reconstructed using a single event camera and 3D-GS. The authors describe a process where the 3D scene reconstruction does not require provided camera poses. The 3D-GS parameters and camera poses are simultaneously calculated, using a concept similar to SLAM, but generating a dense 3D scene.

2) The presented method produces results that outperform the current state-of-the-art by a significant margin.

**Strengths:**

1) An original concept in which 3D-GS and camera poses are optimized simultaneously.

2) Results that surpass the current state-of-the-art.

**Weaknesses:**

The manuscript is clearly written but does not explain in a precise and in-depth manner how it is carried out. In other words, the concepts expressed are only shown at a high level, without delving into small key details, such as the “continuous time trajectory parameterization” or how “the camera poses (T_k) and (T_{k+\Delta t}) can be interpolated,” and how exactly to “render two grayscale images (i.e., (\hat{I}k) and (\hat{I}{k+\Delta t})) from the previously recovered 3D-GS,” which makes it very difficult to reproduce the results.

Although the manuscript mentions some studies related to 3D-GS and event cameras, it does not mention 3D-GS works that perform 3D reconstruction or novel view synthesis with pose-free cameras and frame-based cameras.

**Questions:**

1) Although this type of work is new in the area of event cameras, why is there no mention of other pose-free camera work in the field of frame-based cameras?

2) Since the document only expresses high-level ideas, are there any plans to make the code publicly available in the future?

3) Why are there no supplementary videos supporting the results shown in the manuscript?

**Details Of Ethics Concerns:**

There are no concerns regarding ethics

---

### Official Review · Reviewer_X8Zm · 2024-11-04

**Soundness:** 2
**Presentation:** 3
**Contribution:** 2
**Rating:** 5
**Confidence:** 4

**Summary:**

This paper proposes IncEventGS, an incremental dense 3D reconstruction method using a single event camera. To incrementally recover the 3D scene, IncEventGS leverages the tracking and mapping approach of traditional SLAM. The tracker first estimates initial camera motion from prior 3DGS reconstructions, while the mapper refines both the 3D scene and camera motion using the tracker’s motion trajectory estimates. The advantage of IncEventGS does not require any ground truth camera poses. The results show that IncEventGS outperforms prior NeRF-based methods and related baselines, even without ground-truth camera poses. Additionally, it surpasses SOTA event-based VO methods in camera motion estimation.

**Strengths:**

1. The topic of event-based 3D reconstruction without camera pose is very interesting topic.

2. The authors conducted extensive experiments demonstrating that IncEventGS outperforms previous NeRF-based methods and other baselines, even without ground-truth camera poses.

3. The writing is clear and easy to understand.

**Weaknesses:**

1. While I acknowledge that this paper is the first to explore 3D reconstruction using a single event camera combined with 3D ground segmentation without camera poses, its novelty appears to be limited. There are existing works using traditional RGB cameras for 3D reconstruction without relying on camera poses, and the approach of directly accumulating events into event images does not clearly highlight significant contributions to the field, whether from the image-based 3D ground segmentation community or the event-based community. I encourage the authors to articulate the specific technical contributions of this work.

2. I recommend that the authors include more examples of extreme scenarios, such as high-speed motion and low-light conditions, alongside comparisons with RGB images. This could better demonstrate the advantages of using event cameras for 3D reconstruction.

3. Regarding the possibility of achieving colored 3D reconstruction, can this method be applied? Since there are existing color event cameras, could the authors obtain data from such cameras to create an example of colored reconstruction?

4. The writing could be further improved in several ways: a) The title in line 97 should be bolded and capitalized. b) Section 3.2 does not require an extensive explanation of event camera principles and image accumulation. c) The font sizes in Tables 1 and 2 should be made consistent.

**Questions:**

Please see the weaknesses. I have assigned a preliminary score based on the initial manuscript, but I may adjust this score based on the authors' responses and feedback from other reviewers.

---

### Note · Authors · 2024-11-15

I have read and agree with the venue's withdrawal policy on behalf of myself and my co-authors.